# Oocyte vitrification disrupts zygotic genome activation in embryos by impairing maternal spliceosome translation and *Crxos* splicing

**Jianpeng Qin**[1☯], **Ao Ning**[1☯], **Jian Han**[2☯], **Xiangyi Chen**[1], **Beijia Cao**[1], **Yujun Yao**[2], **Xiaoqing He**[2], **Bo Pan**[1], **Yaozong Wei**[1], **Kunlin Du**[1], **Shuqi Zou**[1], **Jiangfeng Ye**[1], **Guozhi Yu**[3], **Qiuxia Liang**[3], **Jie Qiao**[2], **Jie Yan**[2*], **Guangbin Zhou**[1*]

**1** State Key Laboratory of Swine and Poultry Breeding Industry, Key Laboratory of Livestock and Poultry Multiomics, Ministry of Agriculture and Rural Affairs, Farm Animal Genetic Resources Exploration and Innovation Key Laboratory of Sichuan Province, College of Animal Science and Technology, Sichuan Agricultural University, Chengdu, China, **2** Center for Reproductive Medicine, Department of Obstetrics and Gynecology Peking University Third Hospital, National Clinical Research Center for Obstetrics and Gynecology, State Key Laboratory of Female Fertility Promotion, Key Laboratory of Assisted Reproduction (Peking University), Ministry of Education, Beijing Key Laboratory of Reproductive Endocrinology and Assisted Reproductive Technology, Beijing, China, **3** College of Life Science, Sichuan Agricultural University, Ya'an, China

☯ Contribute equally
\* yanjiebjmu@bjmu.edu.cn (JY), zguangbin@sicau.edu.cn (GZ)

## Abstract

Oocyte vitrification is indispensable in assisted reproduction, yet its link to compromised embryonic development remains mechanistically unresolved. Here, this study demonstrate through integrated transcriptome and translatome analysis that vitrification disrupts maternal mRNA translation—sparing global transcriptional output—in mouse oocytes. This translational perturbation prominently suppresses genes encoding spliceosome components, including *Phf5a*, leading to persistent and widespread alternative splicing defects in subsequent 2-cell embryos. Importantly, aberrant splicing specifically depletes the functional full-length transcript of the essential zygotic genome activation (ZGA) regulator *Crxos* (*Egam1*) while elevating a truncated, non-functional variant (*Egam1*^ΔEXON3). Functional analyses confirm that loss of *Crxos* in 2-cell embryos not only compromises developmental progression but also reduces global transcriptional activity, likely via impaired RNA Pol II recruitment and elongation at ZGA genes. Together, this work delineates a linear pathological cascade triggered by oocyte vitrification, comprising maternal translational suppression, spliceosome impairment, *Crxos* aberrant splicing, impaired ZGA, and developmental compromise, thereby offering a mechanistic basis for refining cryopreservation protocols in reproductive medicine.

**Data availability statement:** The T&T-Seq data, RNA-seq data, Stacc-seq data and CUT & Tag-seq data were deposited in the Genome Sequence Archive (GSA) at the China National Center for Bioinformation under GSA number CRA037097, which is publicly accessible at https://ngdc.cncb.ac.cn/gsa. Previously published RNA-seq data of mouse 2-Cell-like cells and ESCs used in this work were from NCBI GEO accession number GSE166216. Previously published proteomics data of mouse oocytes after vitrification used in this work were from NCBI PubMed accession number 35508090.

**Funding:** This work was supported by the National Key Research and Development Program of China (2021YFD1200403 to GZ, 2022YFC2703000 to JY) and the National Natural Science Foundation of China Major Program (T2293764 to JY). The funders had no role in study design, data collection and analysis, decision to publish, or preparation of the manuscript.

**Competing interests:** The authors have declared that no competing interests exist.

## Author summary

Through integrated transcriptome and translatome analysis, we found that oocyte vitrification specifically disrupts maternal mRNA translation, impairing spliceosome function and causing persistent splicing defects. This induces aberrant splicing of *Crxos*, reducing its functional isoform and compromising zygotic genome activation (ZGA). Our findings provide a mechanistic basis for optimizing cryopreservation protocols in reproductive medicine.

## 1. Introduction

Oocyte cryopreservation is a core technology in human assisted reproduction and genetic resource preservation [1]. Vitrification has become a standard cryopreservation protocol owing to its superior capacity to minimize ice crystal-mediated cellular damage [2]. While clinical outcomes confirm the feasibility of achieving live births from vitrified oocytes [3–5], their developmental competence—notably reflected in blastocyst formation rates—remains substantially lower compared to fresh counterparts [6–11]. This diminished developmental potential (e.g., reduced blastocyst rates) poses a significant barrier to broader clinical implementation. However, the molecular mechanisms underlying cryo-induced oocyte damage, especially disturbances in post-transcriptional regulation beyond classical structural and organellar damage, remain incompletely elucidated.

The successful initiation of early embryonic development critically relies on the precisely orchestrated reservoir of maternal factors accumulated during oocyte growth and maturation [12]. In transcriptionally silent MII oocytes and pre-ZGA embryos, regulatory control of gene expression transitions from transcriptional to translational control [13]. The temporally coordinated translation and degradation of maternal mRNAs, coupled with de novo protein synthesis, collectively drive the maternal-to-zygotic transition (MZT) and initiate ZGA [14–16]. Recent studies indicate that translational activation of specific maternal transcription factors—including OBOX1/2/5/7 [17], TPRXL [18], OTX2 [16], TPD-43 [19], and KLF17 [20]—directly regulates ZGA. Beyond translational control, mRNA alternative splicing represents an additional critical layer governing dynamic gene expression patterns in early embryogenesis. As the first major wave of embryonic transcription, ZGA is accompanied by zygotic splicing activation (ZSA), a process largely mediated by maternally deposited splicing factors [21]. Precise regulation of ZSA is essential for embryonic development; proper alternative splicing of CARM1 in mouse 2-cell embryos critically influences the initial lineage-specification event [22]. Therefore, any disruption to maternal mRNAs translation or spliceosome function is likely to exert profound and enduring effects on subsequent embryogenesis.

Vitrification represents an acute physicochemical stress that inevitably disrupts the exquisitely organized molecular architecture of the oocyte [23,24]. While prior studies have predominantly focused on cryo-induced damage to subcellular

structures—including the cytoskeleton [25], spindle [26,27], mitochondria [28–30], endoplasmic reticulum [31], and cortical granules [32]—as well as oxidative stress [33]. However, recent evidence indicates that oxidative stress can trigger the formation of stress granules, thereby altering mRNA stability and translational dynamics [34–36]. In oocytes, a uniquely specialized cell type, it remains a critical and unresolved question whether—and through what mechanisms—cryopreservation disrupts the translational dynamics of the abundant maternal mRNAs and compromises spliceosome function, thereby impairing the precision of ZGA and alternative splicing.

Therefore, we propose a scientific hypothesis that oocyte vitrification impairs developmental competence by disrupting translational homeostasis of maternal spliceosome-associated mRNAs, thereby compromising spliceosome function, eliciting aberrant alternative splicing and the failure of transcriptional activation at critical developmental windows such as ZGA. To interrogate this hypothesis, we implemented a multi-modal analytical framework in mice, integrating parallel transcriptome and translatome sequencing (T&T-seq), small-scale Tn5-assisted chromatin cleavage sequencing (Stacc-seq), cleavage under targets and tagmentation (CUT&Tag), alongside functional perturbations via Trim-Away, siRNA knockdown (KD), and overexpression (OE). Through systematic examination of vitrified-warmed oocytes and their derived preimplantation embryos, we sought to delineate the molecular cascade linking cryopreservation induced translational dysregulation to embryonic developmental defects. Our study not only provides novel theoretical targets for evaluating and refining current oocyte cryopreservation protocols, but also offers new insights into the precision and vulnerability of post-transcriptional regulatory networks during early embryogenesis.

## 2. Results

### 2.1. Oocyte cryopreservation disrupts maternal mRNA translation

Oocyte vitrification significantly reduced the blastocyst development rate following fertilization (57.29±3.96% vs. 78.18±2.76%), with 17.68% of embryos arrested at the 2-cell stage (Figs 1A, S1A and S1B). To investigate the underlying mechanisms, we performed T&T-seq on vitrified-warmed mouse MII oocytes and their derived preimplantation embryos at multiple developmental stages (Fig 1B and S1 Table). Comparison of the density distributions of raw counts and total raw counts between fresh and vitrification groups revealed overlapping density curves and comparable total raw counts without significant differences (S2A and S2B Fig). These results indicate high consistency in sequencing depth and data distribution between the two groups, allowing direct normalization and downstream analysis based on raw counts. Correlation analyses revealed high reproducibility for both transcriptomic and translatomic datasets (S2C Fig). From MII oocytes to blastocysts, the correlation between transcriptome and translatome initially declined, reaching its lowest point at the 4-cell stage, and subsequently increased (S2D Fig). Upon cryopreservation, the transcriptome–translatome correlation in the vitrification group showed a slight increase at the 2-cell stage and a modest decrease at the MII stage compared with the fresh group, while remaining comparable at the remaining stages. (S2D Fig).

Principal component analysis (PCA) clearly segregated transcriptomic and translatomic profiles across developmental stages, delineating a progressive trajectory from MII oocytes to blastocysts (S2E Fig). Transcriptome data revealed clustering of MII oocytes with zygotes, consistent with transcriptionally silent prior to ZGA (S2E Fig). In contrast, translatome profiles distinctly separated MII oocytes from zygotes, indicating active translational activity during the MZT (S2E Fig). Furthermore, translatome data more clearly distinguished post-ZGA embryonic stages, notably 8-cell embryos and morulae, suggesting that translational dynamics serve as a more precise indicator of developmental progression.

Further analysis demonstrated that cryopreservation immediately altered translational expression profiles in MII oocytes, leading to significant up-regulation of 105 genes and down-regulation of 113 genes (Fig 1C). The most pronounced transcriptional alterations were observed at the 2-cell stage, with 6 genes up-regulated and 166 down-regulated (Fig 1D). Given that changes in translational efficiency directly influence protein synthesis [37], we evaluated the effects of cryopreservation on translational efficiency at both MII and 2-cell stages revealing a significant increase in the vitrification group compared with fresh group (Fig 1E and 1F). In MII oocytes, translationally activated genes were predominantly

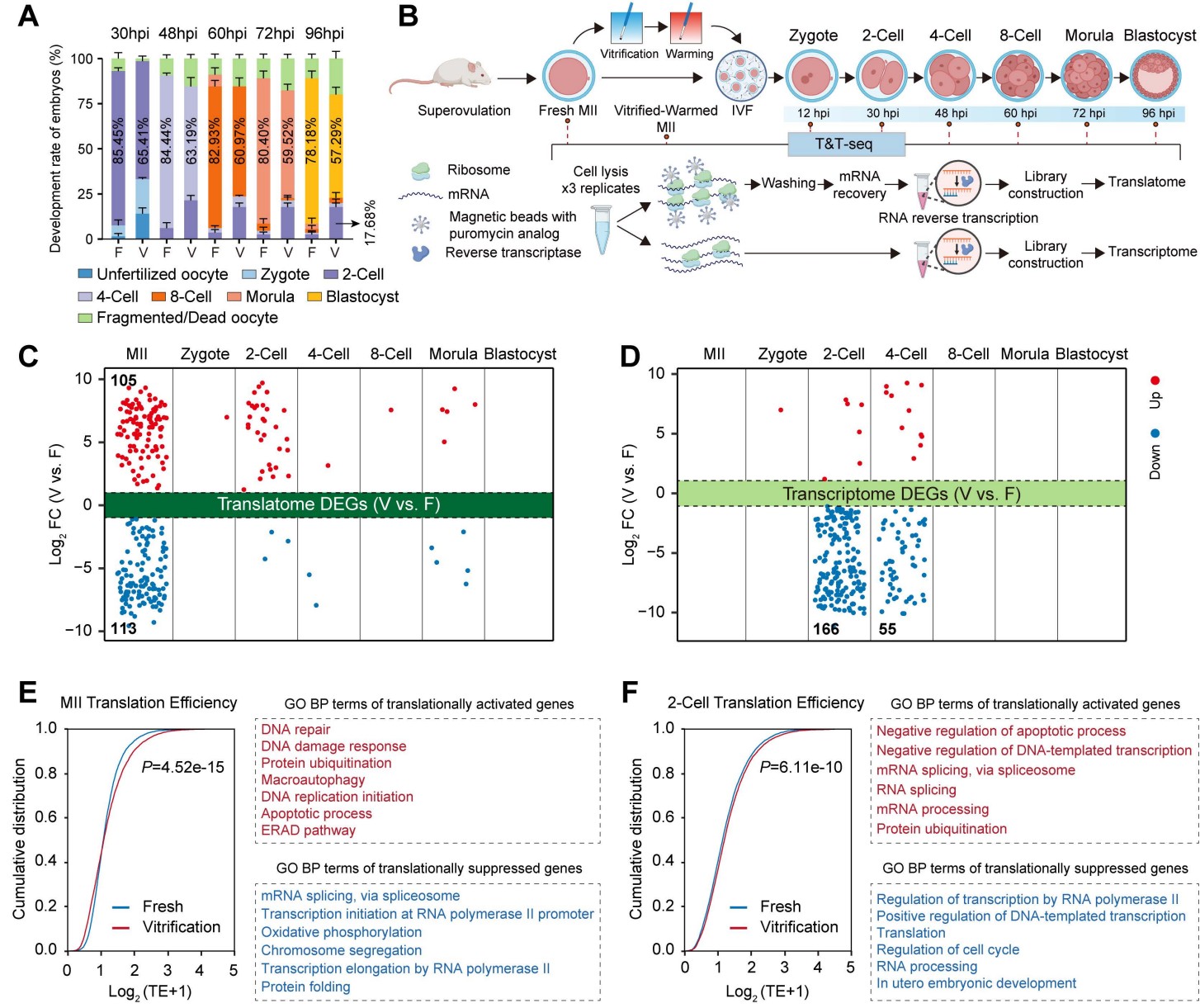

**Fig 1. Impact of oocyte vitrification on the transcriptome and translatome of preimplantation embryos. (A)** Developmental progression of embryos from Fresh (n = 70) and Vitrification (n = 53) groups at 30, 48, 60, 72, and 96 hpi. Stacked bars represent the proportion of embryos at each stage or arrested/fragmented; data are presented as mean ± SEM from three independent experiments. **(B)** Schematic of the T&T-seq workflow applied to oocytes and preimplantation embryos, covering superovulation, MII oocyte vitrification/warming, IVF, and sample collection at 0, 12, 30, 48, 60, 72, and 96 hpi. Created in BioRender. x, **X.** (2026) https://BioRender.com/1ned0v0. **(C, D)** Differentially expressed genes (DEGs) between the Vitrification (V) and Fresh (F) groups across developmental stages. Left: translatome DEGs; right: transcriptome DEGs. Up- and down-regulated genes are colored red and blue, respectively (|log$_2$FC| ≥ 0.58, FDR < 0.05). **(E, F)** TE analysis for MII oocytes (E) and 2-cell embryos (F) in Fresh and Vitrification groups. Enriched biological processes for translationally activated (red) and repressed (blue) genes are shown on the right (P < 0.05).

enriched in pathways associated with DNA repair, protein ubiquitination, autophagy, apoptosis, and endoplasmic reticulum-associated protein degradation (ERAD), whereas translationally suppressed genes primarily involved in mRNA splicing, transcriptional regulation, oxidative phosphorylation, and chromosome segregation (Fig 1E). Unlike MII oocytes,

within 2-cell embryos derived from vitrified oocytes, translationally activated genes were mainly involved in negative transcriptional regulation and RNA splicing, while translationally suppressed genes contributed to positive transcriptional regulation, translation, and cell-cycle progression (Fig 1F). These findings suggest that cryopreservation induced perturbation of maternal mRNA translation may constitute a key mechanism underlying subsequent impairment of ZGA and diminished developmental competence (S2F Fig).

## 2.2. Oocyte cryopreservation impaired ZGA

ZGA is a critical event in early embryonic development, initiating embryonic gene expression and directing subsequent development [38]. Our comparative analysis identified 1354 Major ZGA genes exhibiting consistent expression profiles between fresh and vitrification groups (Fig 2A and S2 Table). However, 346 fresh-specific Major ZGA genes showed lower transcriptional activity in the vitrification group. These genes were functionally enriched in pathways related to mRNA processing, translation, RNA splicing, cell cycle regulation, and RNA Pol II mediated transcription (Fig 2B and 2C). Conversely, 215 vitrification-specific Major ZGA genes exhibited higher transcriptional activity, with predominant involvement in stress-responsive processes such as DNA damage response, repair, and apoptosis (Fig 2B and 2C). Further analysis showed that expression of the ZGA markers MERVL-int and retrotransposon MT2_Mm [39,40] was significantly reduced in 2-cell embryos derived from vitrified oocytes (Fig 2D). EU staining and p-Pol II Ser2 level detection revealed that global transcriptional activity and RNA Pol II elongation capacity were significantly reduced in 2-cell embryos derived from vitrified oocytes (Fig 2E-2H). Collectively, these results indicate that oocyte cryopreservation attenuates transcriptional activity in 2-cell embryos, thereby compromising ZGA.

## 2.3. Oocyte cryopreservation perturbs ZSA

ZGA is accompanied by ZSA. We next analyzed the dynamics of alternative splicing during early embryogenesis. Both fresh and vitrification groups exhibited a large number of alternative splicing events during ZGA, with skipped exons being the predominant form (Fig 3A and S3 Table). The highest number of cryopreservation induced differential alternative splicing events was observed at the 2-cell stage (Fig 3B and S4 Table). Genes displaying differential exon skipping were enriched in functional categories including transcriptional regulation (e.g., *Crxos*, *Setd2*, *Mllt10*), RNA splicing (*Hnrnpa1*), ubiquitination (*Ube3a*, *Ubac1*), apoptosis, autophagy (*Ogt*, *Rmc1*) (Fig 3C and 3D). These findings indicate that oocyte cryopreservation disrupts mRNA alternative splicing in early embryos, potentially affecting subsequent development.

## 2.4. Decreased full-length *Crxos* variant in 2-cell embryos attenuates transcriptional activity

Proper generation of mRNA isoforms through alternative splicing during ZGA is critical for the initiation and sustainability of embryonic development [21,22]. We observed that the transcription and translation levels of *Crxos*, a Major ZGA gene involved in RNA Pol II transcriptional elongation, were unaffected by cryopreservation (S3A-S3C Fig). However, the frequency of exon 3 skipping in *Egam1*, the full-length transcript of *Crxos*, was significantly increased in 2-cell embryos derived from vitrified oocytes (Figs 4A, S3D and S3E), as validated by RT-PCR (Fig 4B and 4C). The two mRNA isoforms differed substantially in coding sequence length and predicted protein structure: full-length Egam1 contained two HOX domains, while Egam1$^{\Delta EXON3}$ lacked these structural domains (S3F and S3G Fig).

To investigate the functional roles of Egam1 and Egam1$^{\Delta EXON3}$, OE in P19 cells demonstrated that *Egam1* encoded a 33 kDa protein, while *Egam1*$^{\Delta EXON3}$ encoded an 11 kDa product (S4A Fig). RNA-seq analysis revealed that OE of *Egam1* markedly influenced gene expression (336 up-regulated, 455 down-regulated) and activated pathways associated with RNA Pol II transcription, cell pluripotency, cell cycle regulation, and ZGA (Figs 4D-4F and S4B-S4D). In contrast, OE of *Egam1*$^{\Delta EXON3}$ exerted minimal effects (39 up-regulated, 26 down-regulated), primarily activating stress-related pathways (Fig 4D-4F). These results confirm that full-length *Crxos* (*Egam1*) possesses transcriptional activation capacity, while *Egam1*$^{\Delta EXON3}$ is functionally deficient in this regard.

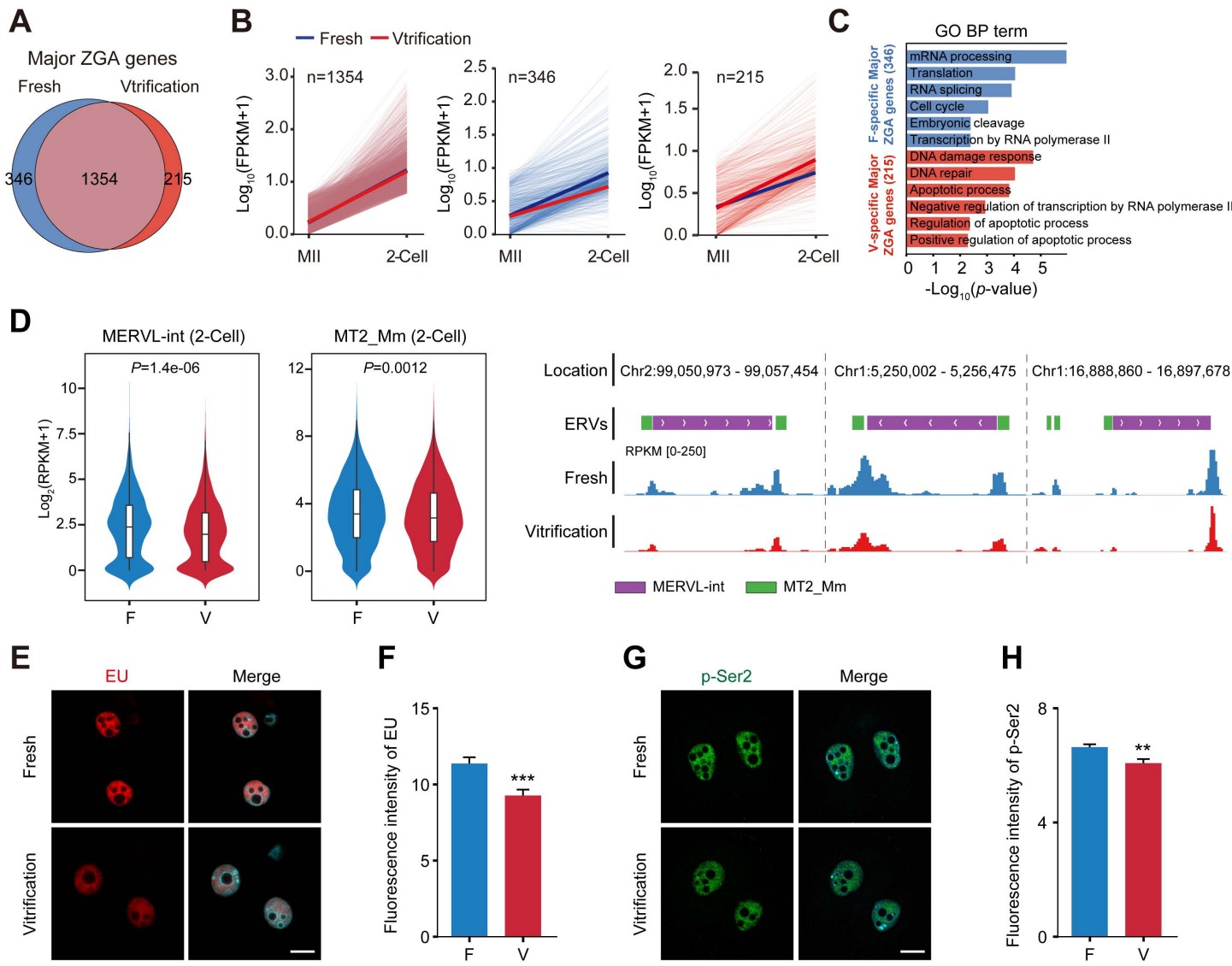

**Fig 2. Oocyte vitrification impairs transcriptional activity in 2-Cell embryos. (A)** Venn diagram showing the overlap of Major ZGA genes between Fresh and Vitrification groups. **(B)** Expression trajectories of the three gene categories from **(A)** (overlap: n = 1354; Fresh-specific: n = 346; Vitrification-specific: n = 215) from the MII to the 2-cell stage. Expression levels are normalized as $\log_{10}$(FPKM + 1). Solid blue and red lines represent median expression in the Fresh and Vitrification groups, respectively. **(C)** GO enrichment of Fresh-specific (F-specific) and Vitrification-specific (V-specific) genes. Blue and red bars denote pathways enriched in the Fresh and Vitrification groups, respectively ($P < 0.05$). **(D)** Expression of endogenous retroviruses (ERV) -based ZGA markers. Left: Violin plots of MERVL-int and MT2_Mm expression levels [$\log_2$(FPKM + 1)] in the two groups. P-values from two-sided Wilcoxon rank-sum test. Right: UCSC Genome Browser tracks showing RNA-seq signal coverage for MERVL-int (purple) and MT2_Mm (green) at specific genomic loci. **(E)** EU incorporation assay visualizing nascent RNA synthesis in 2-cell embryos. EU signal (red) merged with DAPI (cyan). Scale bar = 20 µm. **(F)** Quantification of EU fluorescence intensity in 2-cell embryos from Fresh (n = 30) and Vitrification (n = 31) groups. Data are mean ± SEM from three independent experiments; $***P < 0.001$. **(G)** Immunofluorescence staining of RNA Pol II phosphorylation at Serine 2 (p-Ser2) to assess transcriptional elongation. p-Ser2 (green) is merged with DAPI (cyan). Scale bar = 20 µm. **(H)** Quantification of p-Ser2 signal intensity in 2-cell embryos from Fresh (n = 42) and Vitrification (n = 28) groups. Data are mean ± SEM from three independent experiments; $**P < 0.01$.

Based on these observations, we injected siRNA targeting full-length *Crxos* into early zygotes. This intervention was found to effectively reduce *Crxos* mRNA abundance in 2-cell embryos (S4E Fig) and substantially impaired developmental progression, as evidenced by decreased rates of transition from 2-cell to 4-cell (65.93 ± 9.80% vs. 94.54 ± 2.75%) and

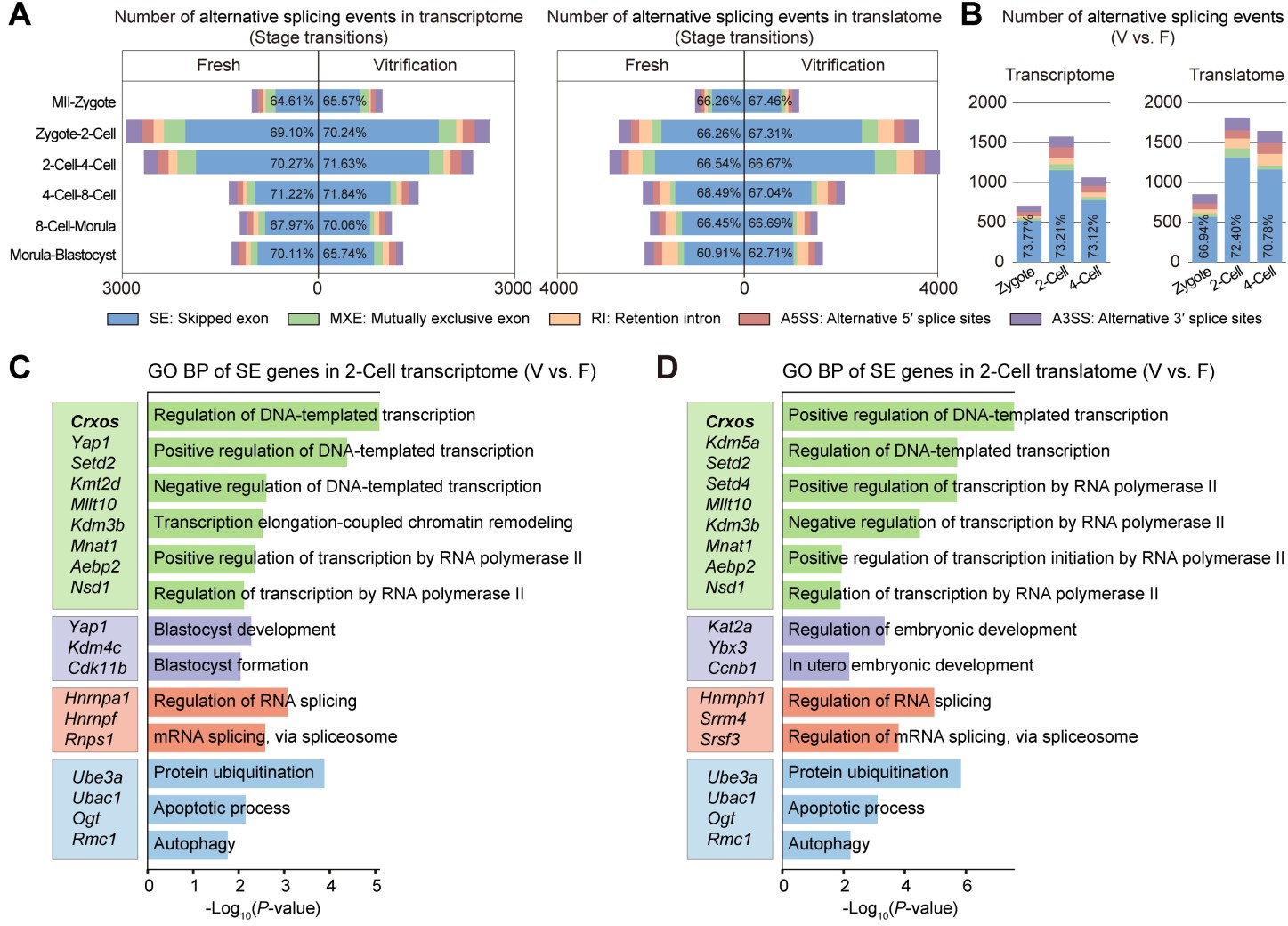

**Fig 3. Oocyte vitrification leads to abnormal alternative splicing in preimplantation embryos. (A)** Alternative splicing events detected by T&T seq during stage transitions. Left: transcriptome; right: translatome. Five alternative splicing types are shown: skipped exon (SE), retained intron (RI), mutually exclusive exons (MXE), alternative 5' splice site (A5SS), and alternative 3' splice site (A3SS). Percentages indicate SE proportion among total events. **(B)** DAS events between Vitrification and Fresh groups at zygote, 2-cell, and 4-cell stages. **(C, D)** GO enrichment of genes with differential SE events in the 2-cell transcriptome (C) and translatome **(D)** (Vitrification vs. Fresh, $P < 0.05$). Representative genes are listed.

blastocyst formation (42.70 ± 11.08% vs. 85.25 ± 3.61%) (Figs 4G, 4H and S4F). Consistent with these phenotypic defects, both EU staining and p-Pol II Ser2 detection demonstrated that *Crxos*-KD significantly attenuated global transcriptional activity and RNA Pol II elongation efficiency in 2-cell embryos (Fig 4I–4L).

## 2.5. Oocyte cryopreservation compromises RNA Pol II pre-configuration and elongation in 2-cell embryos

To comprehensively evaluate how oocyte cryopreservation affects the transcriptional machinery during ZGA, we compared genome-wide RNA Pol II binding profiles in 2-cell embryos derived from fresh and vitrified oocytes using Stacc-seq. In both maternal and Major ZGA gene sets, well-defined Pol II peaks were centered on transcription start sites (TSS) and extended across gene bodies (Fig 5A). Compared with fresh groups, cryopreservation resulted in a widespread attenuation of Pol II signal intensity, which was notably more evident within Major ZGA genes (Fig 5A). Heatmap visualization

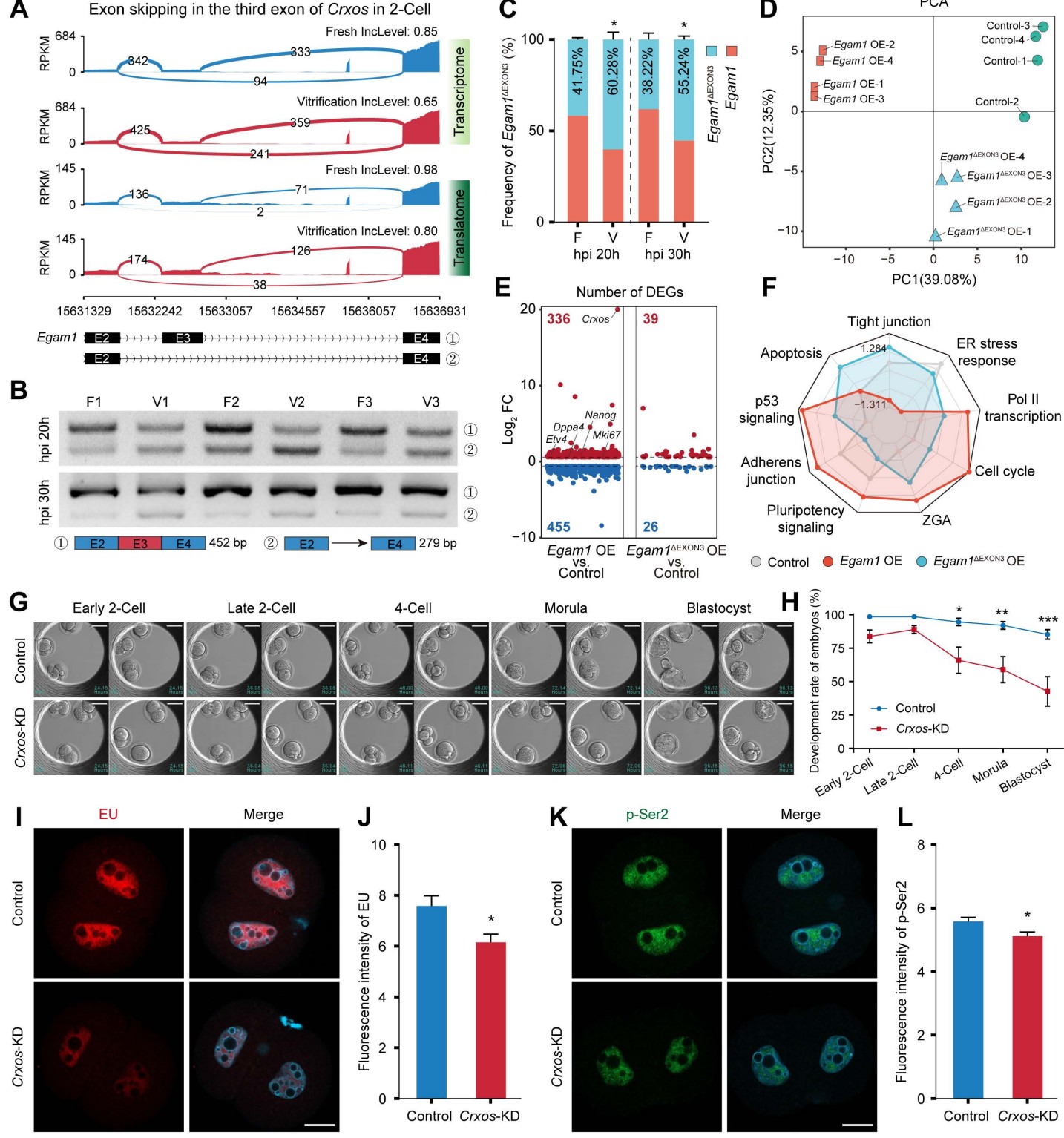

**Fig 4. Functional characterization of different *Crxos* variants. (A)** T&T-seq tracks showing a SE event at the third exon of the *Crxos* (*Egam1*) transcript in 2-cell embryos from vitrification group. Blue and red denote Fresh and Vitrification, respectively. Coverage is normalized by RPKM. Sashimi plots display splice junction read counts and inclusion levels. **(B)** RT-PCR validation of *Crxos* alternative splicing patterns in 2-cell embryos.

Primers flanking Exons 2–4 yield 452 bp (with Exon 3) and 279 bp (without Exon 3) product. Representative of three replicates. **(C)** Quantification of the frequency of exon 3 skipping in the full-length *Crxos* transcript (*Egam1*) from the three replicates shown in **B**. Data are mean±SEM from three independent experiments; *$P<0.05$. **(D)** PCA of RNA-seq data from Control, *Egam1*-OE, and *Egam1*$^{ΔEXON3}$-OE P19 cells. **(E)** DEGs in *Egam1*-OE and *Egam1*$^{ΔEXON3}$-OE cells relative to the Control (|log$_2$FC|>= 0.58, $P<0.05$). Up and down regulated genes are colored red and blue, respectively. **(F)** Radar chart showing pathway enrichment scores across Control, *Egam1*-OE, and *Egam1*$^{ΔEXON3}$-OE groups based on RNA-seq. **(G)** Representative images of embryos at indicated stages in Control and *Crxos*-KD groups. Scale bar=50 μm. **(H)** Developmental rates of Control (n=116) and *Crxos*-KD (n=131) embryos. Data are mean±SEM from three independent experiments; ***$P<0.001$, **$P<0.01$, *$P<0.05$. **(I)** EU incorporation assay for nascent RNA in 2-cell embryos. EU (red) merged with DAPI (cyan). Scale bar=20 μm. **(J)** Quantification of EU fluorescence in Control (n=21) and *Crxos*-KD (n=14) 2-cell embryos. Data are mean±SEM from three independent experiments; *$P<0.05$. **(K)** Immunofluorescence staining of p-Ser2 (green) to assess transcriptional elongation. Merge includes DAPI-stained nuclei (cyan). Scale bar=20 μm. **(L)** Quantification of p-Ser2 signal in Control (n=33) and *Crxos*-KD (n=22) 2-cell embryos. Data are mean±SEM from three independent experiments; *$P<0.05$.

revealed that signal reduction in Major ZGA genes occurred not only near the TSS but also propagated throughout gene bodies and distal intergenic regions in vitrification groups (Fig 5A). Genome wide comparison indicated a broad decrease in Pol II occupancy in the vitrification groups, with a significant reduction across gene body regions (Fig 5B). Moreover, the proportion of Pol II binding sites located within promoter regions (≤3 kb) was higher in vitrification groups, especially within 0–1 kb, suggesting a genome-wide redistribution of RNA Pol II binding sites in 2-cell embryos derived from vitrified oocytes (Fig 5C).

Further analysis showed 946 Pol II binding sites with significantly lower signals in the vitrification group, while only 11 sites showed increased signals (Fig 5D). Motif enrichment analysis of these down-regulated sites revealed significant enrichment for the binding motif of the PRD-like homeodomain transcription factor family—TAATCC [17], which includes factors such as Otx2, GSC, and CRX (Fig 5D). Additionally, these sites were enriched for binding motifs of the pioneer factor Nr5a2 [41] and maternal factor KLF17 [20] (Fig 5D). Given that KD of full-length *Crxos* (*Egam1*) attenuated transcriptional activity in 2-cell embryos (Fig 4H–4K), we performed CUT&Tag on *Egam1*-OE P19 cells to map its genome-wide binding. The resulting Egam1 binding peaks were significantly enriched for the same PRD-like family motif (Fig 5E). Accordingly, we observed significantly reduced Pol II occupancy at the promoter regions of embryo development-related genes *Top2a* [42] and *Prdm10* [43], and the expression levels of these genes were reduced in the vitrification groups. These promoter regions contained both Egam1 binding peaks and the TAATCC motif (Fig 5F). Collectively, these results suggest that the decreased full-length *Crxos* (*Egam1*) in 2-cell embryos derived from vitrified oocytes, likely attenuates RNA Pol II recruitment, ultimately leading to compromised ZGA (Fig 5G).

### 2.6. Reduction of maternal spliceosome protein Phf5a in vitrified oocytes disrupts alternative splicing in 2-cell embryos

To elucidate the cause of aberrant alternative splicing in 2-cell embryos induced by oocyte cryopreservation, we integrated T&T-seq and proteomic data [44] of vitrified-warmed MII oocytes. Analysis revealed four key processes enriched in translationally suppressed genes and downregulated proteins: spliceosome mediated mRNA splicing (Phf5a, Lsm8), oxidative phosphorylation (Cox6b2), negative regulation of apoptosis, and protein transport (Figs 6A and S5A). Given the importance of mRNA alternative splicing for early development [21,22], we focused on the spliceosome protein Phf5a [45,46]. Immunofluorescence and western blot confirmed significantly reduced Phf5a protein levels in both vitrified-warmed MII oocytes and subsequent 2-cell embryos (Fig 6B-6E). Consistent with the T&T-seq data (S5B Fig), these results indicate that vitrification reduces Phf5a protein expression primarily by decreasing its translational efficiency.

To investigate the functional consequences of reduced maternal Phf5a, we employed the Trim-Away specific KD of Phf5a in early zygotes (Fig 6F–6H). While Phf5a-KD did not impair early embryo cleavage (90.97±1.84% vs. 94.44±0.69%), it significantly reduced the rate of 2-cell to 4-cell transition (65.28±5.01% vs. 88.89±3.87%) and blastocyst formation (61.11±1.39% vs. 84.72±3.47%) (Fig 6I and 6J). RNA-seq analysis of Phf5a-KD 2-cell embryos revealed up-regulation of 82 genes associated with transcriptional and cell cycle regulation processes, and down-regulation of 260

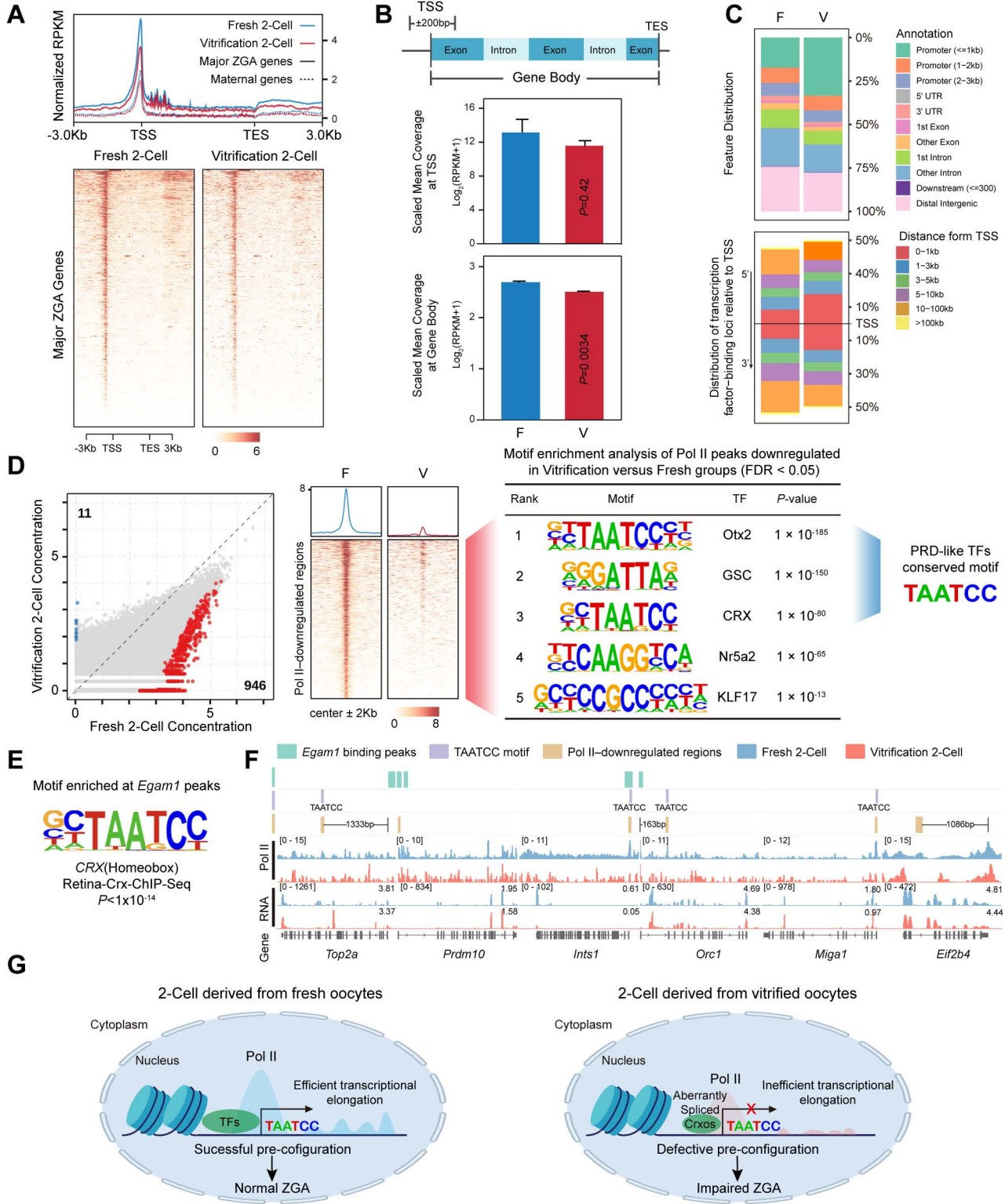

**Fig 5. Oocyte vitrification impairs RNA Pol II pre-configuration and transcriptional elongation in 2-cell embryos. (A)** RNA Pol II occupancy at maternal and Major ZGA genes in 2-cell embryos from Fresh and Vitrified groups. Top: Metaplots showing average Z-score normalized Pol II signals from 3 kb upstream of the TSS to 3 kb downstream of the TES. Bottom: Heatmaps of Pol II signals at Major ZGA genes. **(B)** Quantitative comparison of Pol II signals across genomic features. Top: Schematic of analyzed gene regions. Bottom: Bar charts of average Pol II signal [log$_2$(RPKM + 1)] at the

promoter (TSS +/- 200 bp) and gene body in 2-cell embryos. Two-sided t-test; *P*-values shown. **(C)** Genomic distribution of Pol II signals. Top: Proportion of Pol II signals in functional regions (promoter, UTR, exon, intron, and intergenic). Bottom: Distribution of Pol II signal distance relative to the TSS. **(D)** Differential Pol II signaling and regulatory motifs. Left: Scatter plot of Pol II peak intensity; red dots indicate significantly downregulated signals in the Vitrified group (n = 946). Middle: Heatmap of Pol II signals at downregulated regions (summit ± 2 kb). Right: Motif enrichment analysis of downregulated Pol II regions (*P* < 0.05). **(E)** Motif analysis of Egam1 binding peaks identified by CUT&Tag in *Egam1*-OE P19 cells. **(F)** UCSC Browser tracks of representative genes showing Egam1 binding peaks, TAATCC motif locations, downregulated Pol II regions in the Vitrified group, and Stacc-seq/RNA-seq signals. **(G)** Proposed model of the interaction between Pol II and transcription factors in 2-cell embryos. In fresh embryos, transcription factors recruit Pol II to ensure efficient elongation and normal ZGA. In embryos derived from vitrified oocytes, this process is disrupted: the DNA-binding capacity of PRD-like transcription factors (e.g., aberrantly spliced *Crxos*) is reduced, leading to impaired Pol II recruitment and elongation, which ultimately results in ZGA defects. Created in BioRender. x, **X.** (2026) https://BioRender.com/1ned0v0.

genes associated with ribosome biogenesis, translation initiation, transcriptional regulation, and blastocyst development (S5C-S5E Fig).

As a key component of the U2-snRNP complex [46], Phf5a-KD induced 1072 differential alternative splicing events in 2-cell embryos (Fig 6K and S5 Table). Skipped exon represented the most frequent alteration (68.28%, 732/1072), followed by retained intron, mutually exclusive exon, alternative 5' splice site, and alternative 3' splice site (Fig 6K). Genes with differential splicing events were primarily involved in transcriptional regulation (e.g., *Kdm5a*, *Tead4*, *Yap1*, *TCF3/12*), cell cycle progression (e.g., *Cdk8*, *Cdk16*, *Cdkn1b*, *Ccng2*, *Ccnl2*), and embryonic development (*Mcrs1*) (Fig 6L). Together, these results demonstrate that reduction of the maternal spliceosome protein Phf5a directly disrupts alternative splicing in early embryos.

## 3. Discussion

This study demonstrates that cryopreservation does not significantly alter the oocyte transcriptome, diverging from previous reports that cryopreservation primarily induces widespread transcriptional alterations [47–49]. Instead, the principal outcome of cryopreservation is a widespread disruption of maternal mRNA translation in oocytes. Because translational regulation represents the dominant form of gene expression in transcriptionally silent oocytes and early embryos [12,15,50], this observation highlights the importance of evaluated oocyte cryo-damage at the translational level and provides a new perspective for evaluating the quality of cryopreserved oocytes.

Further analysis of T&T-seq data, we found that oocyte cryopreservation reduces the translational efficiency of spliceosome related genes, including the U2-snRNP subunit Phf5a [46], resulting in a significant decrease in its protein levels in both vitrified-warmed oocytes and subsequent 2-cell embryos. To evaluate the functional impact of this reduction, we employed Trim-Away to KD maternal Phf5a in early zygotes. This intervention recapitulated the cryopreservation phenotype—partial developmental arrest at the 2-cell stage, and accompanied by a large number of genes exhibiting aberrant alternative splicing involved in transcriptional regulation (e.g., *Kdm5a* [51], *Tead4* [52], *TCF3/12* [53,54]) and embryogenesis (e.g., *Mcrs1* [55]). These results directly demonstrate that insufficient maternal spliceosome components contribute importantly to the developmental deficits caused by oocyte cryopreservation. Notably, single-protein KD of Phf5a in our model did not replicate the *Crxos* splicing abnormality observed in 2-cell embryos derived from vitrified oocytes. This difference may stem from the fact that cryopreservation suppresses translation of multiple maternal spliceosome proteins besides Phf5a, such as Sf3b5/6 in U2-snRNP [56,57] and Lsm2/3/5/7/8 in U6-snRNP [58]. Given that pre-mRNA splicing depends on coordinated activities of several snRNPs [59], cryopreservation likely disrupts spliceosome activity more severely than loss of a single factor, leading to more extensive and complex splicing defects.

Among the aberrantly spliced genes, a key finding concerns the transcription factor *Crxos* [60]. While its transcriptional level was unchanged, the functional full-length isoform (*Egam1*) was significantly reduced due to increased skipping of exon 3, whereas a non-functional truncated variant (*Egam1*$^{\Delta EXON3}$) accumulated. Functional studies confirmed that Egam1 activates pathways linked to pluripotency and embryonic development [61], a capacity that is absent in Egam1$^{\Delta EXON3}$. KD

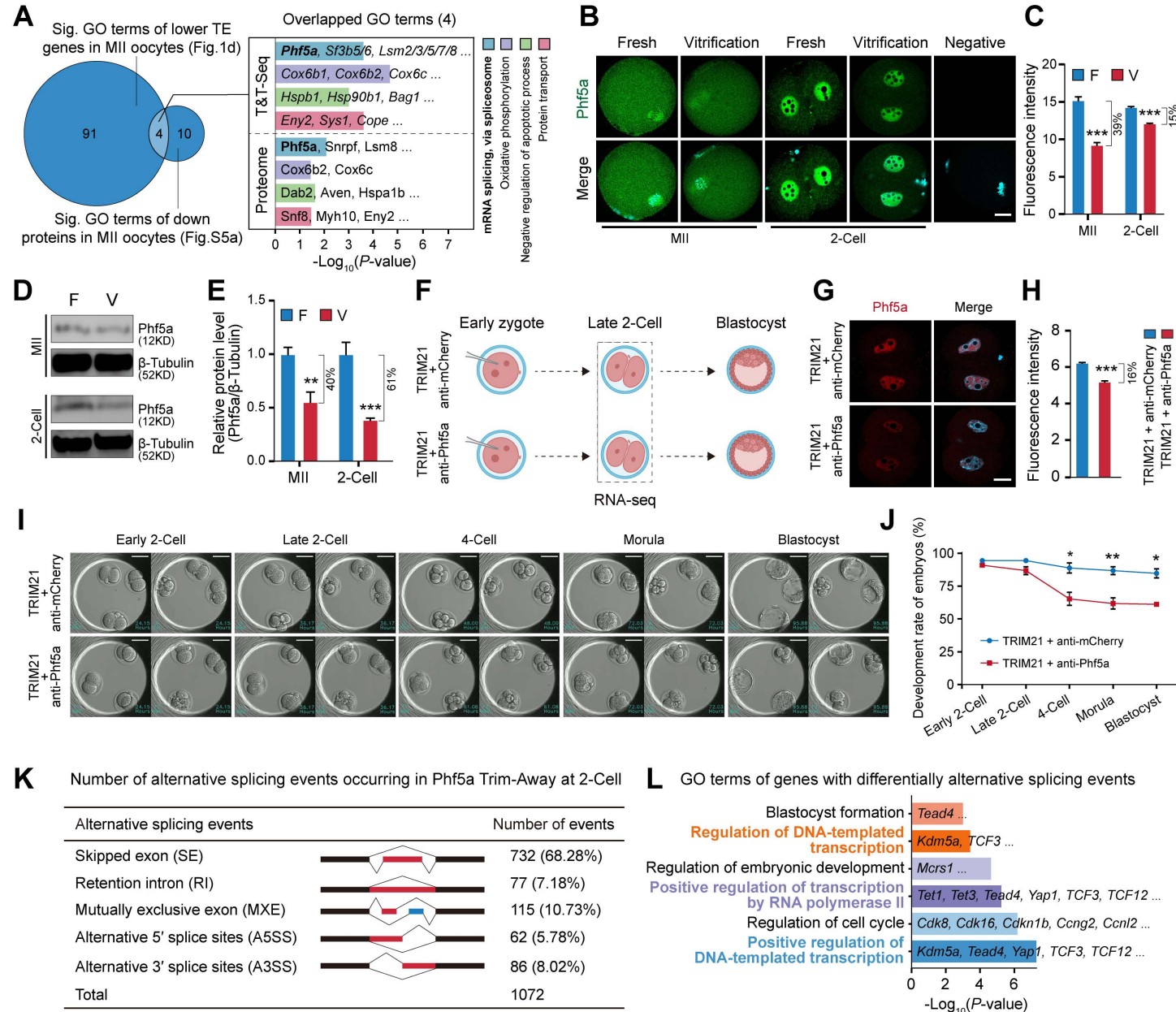

**Fig 6. Maternal Phf5a reduction leads to alternative splicing dysregulation in 2-cell embryos. (A)** Overlap of enriched GO terms between translationally repressed genes and downregulated proteins from vitrified oocytes. Four shared terms are highlighted ($P<0.05$). **(B)** Immunofluorescence staining of Phf5a in MII oocytes and 2-cell embryos. Phf5a (green) merged with DAPI-stained nuclei (cyan). Scale bar = 20 μm. **(C)** Quantification of Phf5a fluorescence intensity in MII oocytes (Fresh, n = 37; Vitrification, n = 39) and 2-cell embryos (Fresh, n = 32; Vitrification, n = 51). Data are mean ± SEM from three independent experiments; ***$P<0.001$. **(D, E)** Western blot and quantitative analysis of Phf5a protein levels in oocytes and 2-cell embryos. β-Tubulin as loading control. Data are mean ± SEM from three independent experiments; ***$P<0.001$, **$P<0.01$. **(F)** Schematic of Phf5a-KD via Trim-Away technology. Created in BioRender. x, **X.** (2026) https://BioRender.com/1ned0v0. **(G, H)** Immunofluorescence staining and quantification of Phf5a in 2-cell embryos after KD. **(I, J)** Representative images and developmental rates of Control (n = 54) and Phf5a-KD (n = 54) embryos. Data are mean ± SEM from three independent experiments; **$P<0.01$, *$P<0.05$. **(K)** DAS events in Phf5a-KD 2-cell embryos. **(L)** GO enrichment of genes with DAS events in Phf5a-KD embryos ($P<0.05$).

of *Crxos* in early zygotes directly reduced global transcriptional activity and RNA Pol II elongation in 2-cell embryos and impeded embryonic development. These data uncover a previously unrecognized mechanism: oocyte cryopreservation can alter alternative splicing of a pivotal transcription factor, drastically curtailing production of its functional protein without changing total mRNA levels, thereby impairing ZGA.

To unravel the genome-wide mechanism underlying defective ZGA, we performed Stacc-seq to profile RNA Pol II binding. The profiles revealed that in 2-cell embryos from vitrified oocytes, Pol II is aberrantly redistributed genome wide, with signals accumulating preferentially near promoters and exhibiting severely impaired elongation into gene bodies. This aberrant "promoter-proximal pausing" is a hallmark of failed transcriptional activation [62,63]. Motif analysis further showed that down-regulated Pol II sites are highly enriched for the TAATCC motif bound by key ZGA transcription factors such as OBOX [17] and OTX2 [16]. Strikingly, OE of full-length *Crxos* (*Egam1*) in P19 cells followed by CUT&Tag indicated that its binding peaks are also enriched for the TAATCC motif. These observations suggest that cryopreservation mediated reduction of full-length *Crxos* likely diminishes its capacity to facilitate effective Pol II pre-configuration and elongation during ZGA, thereby mechanistically coupling splicing defects in an embryonic transcription factor to transcriptional failure.

In summary, our study systematically delineates a cascade initiated by oocyte cryopreservation that leads to embryonic arrest: cryopreservation first disrupts translation of maternal mRNAs, including the core U2-snRNP scaffold Phf5a; this maternal deficit is transmitted to the embryo, causing widespread alternative splicing aberration at the 2-cell stage, notably a decline in the functional full-length isoform of the transcription factor *Crxos*; this reduction likely impairs Pol II pre-configuration and elongation during ZGA, resulting in global decline in transcriptional output and ultimately pre-implantation developmental arrest (Fig 7). This study offers the first evidence of an intergenerational regulatory network connecting "maternal translational suppression" to "embryonic splicing aberration" and finally "transcriptome activation failure". It not only enhances our understanding of gamete stress biology and early embryonic programming but also provides critical theoretical foundations and potential therapeutic targets for optimizing oocyte cryopreservation.

## 4. Materials and methods

**Ethics Statement:** All experimental protocols were approved by the Institutional Animal Care and Use Committee (A2022147) and the Animal Ethics and Welfare Committee of Sichuan Agricultural University (AEWC2016), and were conducted in accordance with the guidelines of the China Association for Laboratory Animal Science.

### 4.1. Animals

ICR mice (8 weeks old) for oocyte vitrification and *in vitro* fertilization (IVF) were purchased from Chengdu Dashuo Experimental Animal Co., Ltd. Mice were acclimatized for two weeks under a controlled 12 h:12 h light dark cycle (lights on at 06:00), 18–25 °C, and 50–70% relative humidity.

ICR mice (8 weeks old) used for zygote microinjection were supplied by the Department of Laboratory Animal Science, Peking University Health Science Center. Mice were maintained under specific pathogen free conditions with a 12 h:12 h light dark cycle, 18–23 °C, and 40–60% relative humidity.

### 4.2. Oocyte and zygote collection

Female mice were primed with 10 IU of PMSG (San-Sheng, China), followed by 10 IU of hCG (San-Sheng, China) 48 h later. Oocytes were harvested at 14 h post-hCG injection. To collect zygotes, hCG-primed females were mated with males for 16 h. At the specified time points, females were euthanized via cervical dislocation to harvest the oviducts. Cumulus-oocyte complexes (COCs) were collected and treated with 300 IU/mL hyaluronidase for 3–5 min to remove cumulus cells. All oocytes and zygotes were washed three times in M2 medium droplets, and those with normal morphology were selected for subsequent experiments.

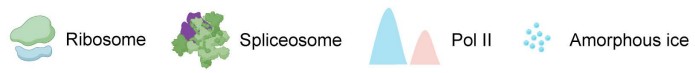

**Fig 7. Graphical Abstract.** Created in BioRender. x, **X.** (2026) https://BioRender.com/1ned0v0.

### 4.3. Oocyte vitrification and warming

The base medium (BM) for vitrification and warming was prepared using HM199 supplemented with 20% FBS. For vitrification, groups of 10 oocytes were picked up using an open-pulled straw (OPS) and equilibrated in BM containing 7.5% EG and 7.5% DMSO for 3 min. Subsequently, oocytes were transferred to a vitrification solution (15% EG, 15% DMSO, and 0.5 mol/L trehalose in BM) for 30 s. Finally, the oocytes were tightly arranged on a Cryotop (Kitazato, Japan) and plunged into liquid nitrogen within 1 min. For warming, vitrified oocytes were sequentially placed in BM containing 1.0, 0.5, and 0.25 mol/L trehalose for 1, 3, and 5 min, respectively. Oocytes were then transferred to BM for 5 min and finally moved to M2 medium in a $CO_2$ incubator for 1 h to recover. All procedures were performed on a 37.5°C-heated stage.

### 4.4. IVF and embryo culture

Sperm were collected from the cauda epididymis of male mice euthanized by cervical dislocation and capacitated for 1 h in HTF medium (AiBei, M1355). The capacitated sperm suspension was added to HTF droplets containing oocytes. After 5 h co-incubation (5 hpi), excess sperm were removed by washing, and zygotes were cultured in KSOM droplets (AiBei, M1435) under standard conditions (37.5 °C, 5% $CO_2$, saturated humidity). Embryo development was recorded at 30, 48, 60, 72, and 96 hpi, with proportions of embryos reaching the 2-cell, 4-cell, 8-cell, morula, and blastocyst stages were calculated relative to the initial number of oocytes. Oocytes and their corresponding IVF derived embryos from fresh and vitrified cohorts were designated the Fresh group and the Vitrification group, respectively.

### 4.5. Plasmid construction and cell transfection

Specific primers were designed to seamlessly clone Egam1/Egam1$^{\Delta EXON3}$ cDNA into a pCDNA3.1(+) vector containing an N-terminal 3×FLAG tag. All plasmids were confirmed by Sanger sequencing. P19 cells were cultured in media containing 89% α-MEM, 10% FBS, and 1% P/S. The Egam1/Egam1$^{\Delta EXON3}$ plasmids were transfected into P19 cells to OE using Lipofectamine 3000 (Invitrogen, L3000001) according to the manufacturer's instructions. Subsequent experiments were performed 48 h post-transfection. Primer sequences for seamless cloning are listed in S6 Table.

### 4.6. Microinjection

Phf5a Trim-Away was performed as previously described [64,65] with minor modifications. Briefly, the eGFP-Trim21 mRNA containing a T7 promoter was linearized and *in vitro* transcribed using the HiScribe T7 ARCA mRNA Kit (with tailing) (New England BioLabs, E2060S). The mRNA was recovered using the MicroElute RNA Clean Up Kit (Omegabiotek, R6247-01). Phf5a (Proteintech, 15554–1-AP) and mCherry (Abcam, EPR20579) antibodies were concentrated using Amicon Ultra 50KDa centrifugal filters at 4°C (Merck Millipore, UFC5050). For injection, the antibodies and eGFP-Trim21 mRNA were prepared in a working solution at final concentrations of 1 mg/mL and 0.2 mg/mL, respectively, with an injection volume of 10 pL.

For *Crxos*-KD, 10 pL of *Crxos* siRNA or Control siRNA (20 μM) was injected into the cytoplasm of zygotes, which were then cultured in KSOM (AiBei, M1435) droplets. Sequences for *Crxos* siRNA and Control siRNA are provided in S7 Table.

All microinjections were performed using an Olympus IX71 inverted microscope equipped with a Narishige micromanipulator system. Early zygotes collected 16 h post-mating were used. Injected zygotes were cultured in a time-lapse incubator (HARIOLAB, Hincubator TL-16R, China) at 37°C, 5% $CO_2$, and saturated humidity. The rates of early 2-cell, late 2-cell, 4-cell, morula, and blastocyst development were recorded at 30, 42, 64, 88, and 102 h post-hCG injection, respectively. All developmental rates were calculated based on the number of cultured zygotes.

### 4.7. PCR

Total cDNA was obtained from 10 embryos at the 2-cell stage per group using the TransScript-Uni Cell to cDNA Synthesis SuperMix for Q-PCR (TransGen Biotech, AC301–01). RT-PCR was performed using 2×Taq PCR MasterMix II (TIANGEN, KT211). The amplified products were analyzed via agarose gel electrophoresis and visualized using a gel imaging system. To

quantify the alternative splicing pattern, the frequency of *Egam1* exon 3 skipping was calculated using densitometric image analysis software by dividing the pixel intensity of the exon 3 skipping band by the combined pixel intensity of both the full-length and exon 3 skipping bands, as described in previous reports [66]. Images from three independent biological replicates were obtained from the same gel. RT-qPCR was conducted using the TransStart Tip Green qPCR SuperMix (TransGen Biotech, AQ601–02-V2) on a QuantStudio 3.0 system (Applied Biosystems, USA). Relative mRNA expression levels were calculated using the $2^{-\Delta\Delta Ct}$ method, with *Gapdh* serving as the internal control for normalization. Primer information is provided in S8 Table.

### 4.8. Immunofluorescence staining

Cell samples were fixed in 4% paraformaldehyde for 20 min and permeabilized in PBS (Phosphate Buffered Saline) containing 1% Triton X-100 for 20 min at room temperature. After blocking with 1% BSA in PBS for 1 h, cells were incubated with primary antibodies (S9 Table) overnight at 4°C. Following three 10-min washes in PBS containing 0.01% Triton X-100 and 0.1% Tween 20, cells were incubated with fluorescence-labeled secondary antibodies (S10 Table) for 1 h at 37°C. After three additional washes, samples were mounted on slides with DAPI-containing anti-fade mounting medium (Vector, H-1200) and imaged using a confocal microscope (Zeiss, LSM980). Fluorescence intensity of the target protein signal was quantified using ImageJ software (v1.48, Bethesda, USA). Immunofluorescence procedures and imaging parameters were kept consistent across all groups.

### 4.9. 5-Ethynyl Uridine (EU) Staining

Nascent transcripts in 2-cell embryos were detected using an EU Detection Kit (Ruibo Biotechnology, C10316-1) following the manufacturer's instructions. Embryos were mounted on slides with DAPI-containing anti-fade medium and imaged via confocal microscopy (Zeiss, LSM980). The fluorescence intensity of the EU signal was quantified using ImageJ software (v1.48, Bethesda, USA).

### 4.10. Western blotting

Samples of transfected P19 cells, as well as 200 oocytes or 2-cell embryos per group, were collected and lysed on ice for 10 min in RIPA buffer (BOSTER, AR0102) supplemented with protease inhibitors (BOSTER, AR1182–1) and phosphatase inhibitors (BOSTER, AR1183). Samples were then mixed with 5×SDS-PAGE loading buffer and boiled at 95°C for 10 min. Proteins were separated by SDS-PAGE (Bio-Rad) and transferred to PVDF membranes (EpiZyme, China). After blocking for 2 h at room temperature, membranes were incubated with primary antibodies (S9 Table) overnight at 4°C. After three washes in TBST, membranes were incubated with HRP-conjugated secondary antibodies (S10 Table) for 1 h at room temperature. Protein bands were visualized using ECL (36208ES60, Yeasen) on a western blot detection system. Protein expression was quantified using ImageJ software (v1.48, Bethesda, USA).

### 4.11. T&T-seq of oocytes and preimplantation embryos

The T&T-seq protocol was based on previously reported methods [67,68]. Briefly, 10 oocytes or embryos were treated with acidic Tyrode's solution (Sigma, T1788) for 30 s to remove the zona pellucida, washed three times in PBS, and lysed in 10 µL of lysis buffer (Vazyme, N712) for 10 min on ice. Two microliters of the lysate were used for transcriptome sequencing, while the remaining 8 µL were used for translatome analysis. Translatome mRNA isolation was performed using the RiboLace Beads Kit (Immagina, RL001), followed by purification with the VAHTS RNA Clean Beads Kit (Vazyme, N412). Both transcriptome and translatome mRNAs were reverse-transcribed and amplified using the Single Cell Full Length mRNA-Amplification Kit (Vazyme, N712). The amplified cDNA was purified (VAHTS DNA Clean Beads Kit, Vazyme, N411) and used for library construction using the TruePrep DNA Library Prep Kit V2 for Illumina (Vazyme, TD502/503). Sequencing was performed on the Illumina NovaSeq 6000 platform.

 

### 4.12. RNA-seq of P19 cells

Total RNA was extracted from P19 cells. mRNA was enriched using Oligo(dT) magnetic beads and reverse-transcribed into first-strand cDNA, followed by PCR amplification, product purification, and library construction. Sequencing was performed on the Illumina NovaSeq 6000 platform.

### 4.13. T&T-seq and RNA-seq data processing

Raw data were trimmed and quality-controlled using Trim Galore (v0.6.10). Filtered reads were aligned to the mouse reference genome (GRCm39) using HISAT2 (v2.2.1). SAM files were converted to BAM format, sorted, and indexed using SAMtools (v1.21). Gene expression levels were quantified using StringTie (v3.0.0) and expressed as fragments per kilobase per million mapped reads (FPKM).

### 4.14. Stacc-seq

Stacc-seq was performed as previously described [17] with minor modifications. Briefly, 150 zona-free 2-cell embryos, 1.5 μL pA/G-Tnp (Vazyme, TD904), and 0.5 μg RNA Pol II antibody (Active Motif, 39497) were used to collect antibody-bound DNA. Libraries were amplified using the TruePrep Amplify Enzyme kit (Vazyme, TD601) and sequenced on the Illumina NovaSeq 6000 platform.

### 4.15. CUT&Tag-seq

CUT&Tag libraries were constructed following the protocol by Zou et al.[18] with minor modifications using the Hyperactive Universal CUT&Tag Assay Kit for Illumina (Vazyme, TD904). Briefly, $5 \times 10^4$ *Egam1*-OE P19 cells were immobilized on ConA-coated magnetic beads. After permeabilization, cells were incubated sequentially with FLAG antibody (CST, 14793S) and Goat Anti-Rabbit IgG H&L (Vazyme, Ab207–01), followed by incubation with pA/G-Tnp. Targeted tagmentation was performed to fragment DNA and introduce adapters. Libraries were then amplified, purified, and sequenced.

### 4.16. Stacc-seq and CUT&Tag-seq data processing

Data processing was performed as previously described [17,20]. Raw reads were quality-controlled and trimmed using Trim Galore (v0.6.10) and aligned to the mouse reference genome (GRCm38) using Bowtie2 (v2.5.4). After filtering for mapping quality (MAPQ ≥ 20), removing PCR duplicates, and indexing with SAMtools (v1.21), bigWig files were generated via deepTools (v3.5.6) for visualization in IGV. To minimize batch effects, RPKM values for Stacc-seq were further normalized via Z-score transformation. Signal matrices and profiles were generated using deepTools. Peak annotation, comparison, and visualization were conducted in R using ChIPseeker (v1.44.0). Differential binding sites were identified using DiffBind (v3.14.0) with an FDR < 0.05. Peak calling for both Stacc-seq and CUT&Tag was performed using MACS2 (v2.2.9.1).

### 4.17. Bioinformatics analysis

Translation efficiency (TE) was calculated as the ratio of translatome to transcriptome expression levels (FPKM + 1 / FPKM + 1) following the method described by Xiong et al.[50]. Genes with a TE ratio (vitrified/fresh) >1.5 were defined as translationally activated, whereas those with a ratio <0.67 were defined as translationally suppressed.

Maternal and major ZGA genes were defined based on the criteria from Ji et al.[17]. Maternal genes were defined as those expressed in MII oocytes (FPKM > 5) but significantly downregulated (at least 3-fold) in late 2-cell embryos. Major ZGA genes were defined as genes with low or no expression in MII oocytes (FPKM < 5) that were significantly upregulated (FPKM > 5, at least 3-fold) in 2-cell embryos.

Differential expression analysis was performed using DESeq2 (v1.42.0) or edgeR (v4.0.16). Genes with an absolute $\log_2$ fold change ($|\log_2FC|$) ≥ 0.58 and an adjusted $P$-value (P.adj) or false discovery rate (FDR) < 0.05 were considered significantly upregulated or downregulated.

Alternative splicing events were identified using rMATS package [69] and classified into five categories: skipped exon (SE), retained intron (RI), mutually exclusive exons (MXE), alternative 5' splice site (A5SS), and alternative 3' splice site (A3SS). Differential alternative splicing (DAS) events were defined as those with an FDR < 0.05 and an absolute inclusion level difference (|IncLevelDifference|) > 0.1. Visualization of AS events was performed using the rmats2sashimiplot (v3.0.0).

Motif enrichment analysis was conducted using the findMotifsGenome.pl script in the HOMER suite [70]. Input regions consisted of the ± 200 bp flanking sequences of the downregulated sites identified by Stacc-seq and the peak regions identified from the CUT&Tag.

The expression of transposable elements was analyzed as previously described [71]. Briefly, a mouse transposable elements annotation was constructed based on RepeatMasker and GENCODE mm10, excluding retained introns and regions overlapping with coding exons. T&T-seq reads were mapped to the mouse genome, and transposable elements counts were quantified using featureCounts. Merged bigWig files were generated for visualization. Statistical differences between groups were assessed using the two-tailed Wilcoxon rank-sum test.

The native structures of Egam1 and Egam1$^{\Delta EXON3}$ proteins were predicted using AlphaFold3 [72]. Functional domains within the protein structures were identified and scored via the SMART database [73], using an E-value < 0.01 as the significance threshold. Key functional domains were then highlighted and displayed within the predicted 3D models.

Gene Ontology (GO) and functional enrichment analyses were performed using the DAVID database (https://david.ncifcrf.gov/).

### 4.18. Statistical analysis

Statistical analyses were performed using SPSS software (v20.0; IBM, USA). Quantitative data were analyzed using two-tailed Student's t-tests, with $P < 0.05$ considered statistically significant. All data are presented as mean ± standard error of the mean (SEM). Percentage data were subjected to arcsine transformation prior to statistical analysis to ensure normality. Experimental flowcharts and graphical abstracts were created in BioRender. x, X. (2026) https://BioRender.com/1ned0v0.

## Supporting information

**S1 Fig. Embryonic development after oocyte vitrification/warming.** (A) Representative images of embryos at each stage post-fertilization. Scale bar = 50 μm. (B) Developmental rates of Fresh (n = 70) and Vitrification (n = 53) groups. Data are mean ± SEM from three independent experiments; *$P < 0.05$, **$P < 0.01$.
(TIF)

**S2 Fig. T&T-seq analysis of oocytes and preimplantation embryos.** (A) Density distribution of raw counts between fresh and vitrified groups from T&T-seq data. (B) Bar chart of total raw counts between fresh and vitrified groups from T&T-seq data. (C) Heatmap of pearson correlation coefficients for the transcriptome and translatome. (D) Changes in transcriptome-translatome correlations induced by vitrification. (E) PCA of transcriptome and translatome. (F) Schematic summarizing the patterns of gene expression changes following vitrification. Created in BioRender. x, X. (2026) https://BioRender.com/1ned0v0.
(TIF)

**S3 Fig. Expression profile of *Crxos*.** (A, B) Transcriptional and translational expression levels of *Crxos* across stages. (C) RT-qPCR of *Crxos* expression from 12 to 30 hpi (2 h intervals). (D) UCSC Browser tracks of *Crxos* isoforms in oocytes, embryos, 2C-like cells, and ESCs. (E) RT-qPCR of *Egam1* and *Egam1N* transcripts in 2-cell and blastocyst

stages. (F) PCR amplification of *Egam1* and *Egam1*<sup>ΔEXON3</sup> CDS regions. (G) Predicted protein structures of Egam1 and Egam1<sup>ΔEXON3</sup>.
(TIF)

**S4 Fig. Functional study of *Egam1* and *Egam1*<sup>ΔEXON3</sup>.** (A) Egam1 and Egam1<sup>ΔEXON3</sup> protein expression after OE in P19 cells. (B) Transcriptional impact and GO enrichment of Egam1-activated genes. (C) Venn diagram showing the overlap between Egam1-activated genes and ZGA genes. (D) Heatmap of representative Egam1-activated genes. (E) RT-qPCR validation of *Crxos*-KD efficiency. (F) Stacked bar chart showing embryonic development after *Crxos*-KD.
(TIF)

**S5 Fig. Impact of Phf5a-KD on 2-cell gene expression.** (A) GO enrichment of differentially expressed proteins (DEPs) in vitrified oocytes [44]. (B) *Phf5a* expression was examined by T&T-seq. Translation efficiency was calculated as the ratio of translatome to transcriptome expression levels. Data are mean ± SEM from three independent experiments; $**P<0.01$, $*P<0.05$. (C) MA plot showing DEGs in 2-cell embryos after Phf5a-KD. (D, E) GO enrichment of upregulated (red) and downregulated (blue) genes in Phf5a-KD embryos.
(TIF)

**S1 Table. T&T-seq data from vitrified-warmed MII oocytes and their derived preimplantation embryos.**
(XLSX)

**S2 Table. Major ZGA genes from the fresh and vitrification groups.**
(XLSX)

**S3 Table. Differential alternative splicing events during oocyte and early embryonic development from T&T-seq data.**
(XLSX)

**S4 Table. Differential alternative splicing events between the fresh and vitrification groups during ZGA.**
(XLSX)

**S5 Table. Differential alternative splicing events upon Phf5a Trim-Away at the 2-cell stage.**
(XLSX)

**S6 Table. The primers sequences for seamless cloning.**
(XLSX)

**S7 Table. The siRNA sequences.**
(XLSX)

**S8 Table. The PCR primers.**
(XLSX)

**S9 Table. The information of primary antibodies.**
(XLSX)

**S10 Table. The information of secondary antibodies.**
(XLSX)

## Acknowledgments

The authors would like to thank Dr. Wenqi Hu (California Institute of Technology, California, Pasadena CA91125, America) for providing the protocol of T&T-seq.

## Author contributions

**Conceptualization:** Jianpeng Qin, Ao Ning, Jian Han, Guozhi Yu, Qiuxia Liang, Jie Yan, Guangbin Zhou.

**Data curation:** Jianpeng Qin, Ao Ning, Jian Han.

**Formal analysis:** Jianpeng Qin, Ao Ning, Bo Pan, Yaozong Wei.

**Funding acquisition:** Jie Yan, Guangbin Zhou.

**Investigation:** Jianpeng Qin, Ao Ning, Xiangyi Chen, Beijia Cao, Xiaoqing He, Bo Pan, Yaozong Wei, Kunlin Du, Shuqi Zou.

**Methodology:** Jianpeng Qin, Ao Ning, Jian Han, Guozhi Yu, Qiuxia Liang, Jie Yan, Guangbin Zhou.

**Project administration:** Jiangfeng Ye.

**Resources:** Jie Qiao.

**Validation:** Jianpeng Qin, Ao Ning, Jian Han, Yujun Yao.

**Visualization:** Jianpeng Qin, Ao Ning.

**Writing – original draft:** Jianpeng Qin, Ao Ning, Jian Han.

**Writing – review & editing:** Jie Yan, Guangbin Zhou.

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
