## [Decision Letter · Decision Letter 0]

15 Mar 2026

PGENETICS-D-26-00149

Oocyte Vitrification Disrupts Zygotic Genome Activation in Embryos by Impairing Maternal Spliceosome Translation and Crxos Splicing

PLOS Genetics

Dear Dr. Zhou,

Thank you for submitting your manuscript to PLOS Genetics. After careful consideration, we feel that it has merit but does not fully meet PLOS Genetics's publication criteria as it currently stands. Therefore, we invite you to submit a revised version of the manuscript that addresses the points raised during the review process.

Please submit your revised manuscript within by Apr 14 2026 11:59PM. If you will need more time than this to complete your revisions, please reply to this message or contact the journal office at plosgenetics@plos.org. Please include the following items when submitting your revised manuscript:

We look forward to receiving your revised manuscript.

Kind regards,

Wei-Hsiang Huang

Academic Editor

PLOS Genetics

Pablo Wappner

Section Editor

PLOS Genetics

Aimée Dudley

Editor-in-Chief

PLOS Genetics

Anne Goriely

Editor-in-Chief

PLOS Genetics

**Journal Requirements:**

If you are using the free assets from BioRender, we are unable to publish these images as they are licenced under a stricter licence than CC BY 4.0. In this case we ask you to remove the BioRender images and replace them with open source alternatives.

See these open source resources you may use to replace images / clip-art:

- https://bioart.niaid.nih.gov/

- https://bioicons.com/

- https://healthicons.org/

- https://scidraw.io/

- https://reactome.org/icon-lib

- https://www.phylopic.org/images

- https://journals.plos.org/plosbiology/article?id=10.1371/journal.pbio.3002395

**Reviewers' comments:**

Reviewer #1: Overall Impressions

This study provides a compelling mechanistic link between oocyte vitrification and compromised embryonic development through integrated transcriptome and translatome analyses in mouse oocytes. The authors demonstrate that vitrification specifically disrupts maternal mRNA translation—without affecting global transcriptional output—leading to suppressed expression of spliceosome components, including Phf5a. This perturbation results in persistent alternative splicing defects in 2-cell embryos, notably affecting the zygotic genome activation (ZGA) regulator Crxos (Egam1), where the functional full-length transcript is depleted while a truncated non-coding variant is elevated. Functional assays confirm that Crxos loss in 2-cell embryos impairs both developmental progression and global transcriptional activity, likely through defective RNA Pol II recruitment and elongation at ZGA genes. These findings not only advance our fundamental understanding of post-vitrification embryonic defects but also carry significant translational implications for optimizing cryopreservation protocols in reproductive medicine. The data are robust and well-supported, and the manuscript warrants acceptance after minor revisions.

General Suggestions

1. Please carefully review the manuscript for grammatical issues and improve the clarity of the language throughout the text.

2. Please ensure that all gene names are italicized consistently throughout the manuscript.

3. Please ensure that all figure legends are comprehensive yet concisely presented.

Minor Suggestions and Grammar Corrections

Author Summary

Line 48-50: The sentence structure is slightly awkward. Consider revising to: "Our findings provide a mechanistic basis for optimizing cryopreservation protocols in reproductive medicine."

Introduction

Line 53: The word "dominant" is somewhat subjective and imprecise. Please replace with "a standard" to more objectively reflect its clinical status.

Line 58: Please revise "This developmental compromise poses a significant barrier..." to "This diminished developmental potential (e.g., reduced blastocyst rates) poses a significant barrier..." for greater specificity and impact.

Line 77: Please replace "maternal mRNA" with "maternal mRNAs".

Line 93: Please replace "spliceosome associated mRNAs" with "spliceosome-associated mRNAs".

Line 94: Please replace "failed transcriptional activation" with "the failure of transcriptional activation".

Line 101: Please replace "seek" with "sought".

Results

Line 117: Please replace "before subsequently increasing" with "and subsequently increased".

Line 118: Please replace "fresh group" with "the fresh group".

Line 150: Please replace "subsequent developmental" with "subsequent development".

Line 157: Please replace "stress responsive processes" with "stress-responsive processes".

Line 169-170: Please replace "with skipped exon being the predominant form" with "with skipped exons being the predominant form".

Line 174: Remove the comma before "and".

Line 183-184: Please replace "Crxos's full-length transcript" with "the full-length transcript of Crxos".

Line 185: Please replace "as further validated by PCR" with "as validated by PCR".

Line 196: Please replace "stress related pathways" with "stress-related pathways".

Line 200: Please replace "effectively reduced" with "was found to effectively reduce".

Line 202: Please replace "decreased rates of 2 cell to 4 cell transition" with "decreased rates of transition from 2-cell to 4-cell".

Line 210-211: Please replace "genome wide RNA Pol II binding profiles" with "genome-wide RNA Pol II binding profiles".

Line 212: Please replace "well defined Pol II peaks" with "well-defined Pol II peaks".

Line 245: Please replace "enriched among" with "enriched in".

Line 269-270: Please replace "reduction of maternal spliceosome protein directly disrupts alternative splicing" with "reduction of the maternal spliceosome protein Phf5a directly disrupts alternative splicing".

Discussion

Line 286-287: Please replace "a large number of aberrant alternative splicing genes" with "a large number of genes exhibiting aberrant alternative splicing".

Line 310: Please replace "genome wide mechanism" with "genome-wide mechanism".

Materials and Methods

Line 354-355: Please replace "oocytes were harvested 14 h post-hCG injection" with "oocytes were harvested at 14 h post-hCG injection".

Line 371: Please replace "37.5°C heated stage" with "37.5°C-heated stage".

Reviewer #2: In this manuscript, Qin et al. describe analyses of the effects of oocyte vitrification on gene expression prior to and following zygotic genome activation, to try to identify pathological effects that could contribute to reported reduced IVF success resulting from the vitrification process. They note that indeed oocyte vitrification reduces the rate of developmental progression to blastocyst stage, consistent with prior reports. The authors perform transcriptomic and translatomic measurements to identify potential changes to the gene expression machinery. These experiments can be challenging to interpret if there are global effects, and in fact the raw numbers of differentially expressed genes (DEGs) initially appear to be quite small considering the thousands of genes expressed prior to and post-ZGA. However, the authors do identify some important differentially translated genes such as Phf5a, an important splicing factor, and provide some compelling analyses that suggest a causal role for vitrification in embryonic splicing-associated defects. The overall research question is important given the lack of knowledge of manipulations such as vitrification on oocyte and early embryonic gene expression, and the authors provide important data on potential gene expression defects with important new genomic datasets that may provide quite useful to the field that I believe make this story worthy of publication in PLOS Genetics.

Major points:

“ …the vitrification group showed higher transcriptome-translatome correlation in MII oocytes and 8-cell embryos but lower correlation in at the 2-cell and blastocyst stages (S2B Fig.)

- The differences look very subtle in this figure, given a lack of statistical analysis controlling for multiple testing across various timepoint comparisons it is not apparent if this is a biologically meaningful result.

- If translation utilizing maternally supplied ribosomes/mRNAs was disrupted with consequential effects on ZGA why were there no DEG’s in transcriptome comparisons in post-ZGA stages (8-cell/morula/blastocyst)? It would be helpful to add percentages of genes that were unaffected vs. differential - e.g. out of 12,000 genes detected at the level of transcript abundance and ribosome-association, 1.8% or 218 were significantly differentially expressed. Even at the timepoints with the most DEGs, it would appear that the vast majority of genes were identically expressed between fresh vs. vitrified samples. Why would the effects of vitrification lead to alterations in such a small subset of transcripts? It would seem to me that the fact that the vast majority of genes were not differentially expressed at the level of transcript abundance/translation would support the null hypothesis that the gene expression apparatus of the oocyte/early embryo is largely unaffected by vitrification? Or are only a small fraction of highly expressed genes detected in these analyses, potentially leading to an underestimation of the effects on gene expression?

- Any sense whether global levels of translation were affected? Spike-in or other controls for global effects on gene expression can frequently unmask deficits in translation or RNA stability occurring across the transcriptome that may be otherwise obscured by common normalizations used in genomic analyses (e.g. TPM normalization). How would the analyses changed if accounting for bulk changes in genome-wide transcription as described in Fig. 2E?

- Supplementary tables should be provided with a list of the specific DEG’s described in key figures, such as Fig. 2A.

- In general there are issues with low figure resolution. Figures should be exported at a much higher resolution. For example, Fig. 4C and 4D, 5B etc. are very difficult to read.

- Authors should quantify the RT-PCR experiment shown in Fig. 4B with a statistical analysis of the spliced vs. unspliced form from analyses of the three replicates

- To help interpret the results it would be helpful to be able to compare the decrease for proteins such as Phf5a as measured by T&T-seq vs. the Western blotting by providing additional numbers, e.g. for each protein measured how much was it decreased in this dataset at the levels of translation vs. transcript abundance vs. the net effects on protein levels? This would also help disentangle changes to protein stability vs. synthesis.

- How much knockdown of Phf5a was achieved? It looks like the knockdown was quite subtle in Fig. 6H – is it known if this gene is dosage sensitive where a small degree of knockdown would be predicted to be biologically significant?

**Have all data underlying the figures and results presented in the manuscript been provided?**

Large-scale datasets should be made available via a public repository as described in the *PLOS Genetics*
data availability policy, and numerical data that underlies graphs or summary statistics should be provided in spreadsheet form as supporting information., and numerical data that underlies graphs or summary statistics should be provided in spreadsheet form as supporting information., and numerical data that underlies graphs or summary statistics should be provided in spreadsheet form as supporting information., and numerical data that underlies graphs or summary statistics should be provided in spreadsheet form as supporting information.

Reviewer #1: Yes

Reviewer #2: Yes

PLOS authors have the option to publish the peer review history of their article (what does this mean?). If published, this will include your full peer review and any attached files.). If published, this will include your full peer review and any attached files.). If published, this will include your full peer review and any attached files.). If published, this will include your full peer review and any attached files.

...

Reviewer #1: No

Reviewer #2: No

**Figure resubmission:**
---

## [Decision Letter · Decision Letter 1]

6 Apr 2026

Dear Dr Zhou,

We are pleased to inform you that your manuscript entitled "Oocyte vitrification disrupts zygotic genome activation in embryos by impairing maternal spliceosome translation and Crxos  splicing" has been editorially accepted for publication in PLOS Genetics. Congratulations!

Yours sincerely,

Wei-Hsiang Huang

Academic Editor

PLOS Genetics

Pablo Wappner

Section Editor

PLOS Genetics

Aimée Dudley

Editor-in-Chief

PLOS Genetics

Anne Goriely

Editor-in-Chief

PLOS Genetics

BlueSky: @plos.bsky.social

Comments from the reviewers (if applicable):

Reviewer's Responses to Questions

**Comments to the Authors:**

Reviewer #1: I have reviewed the authors' responses to the previous comments and the corresponding revisions. The issues I raised have been well addressed, and the presentation of the manuscript has significantly improved. I agree that the article can be accepted by PLOS Genetics

Reviewer #2: I am satisfied with the authors' responses to my points

**Have all data underlying the figures and results presented in the manuscript been provided?**

Large-scale datasets should be made available via a public repository as described in the *PLOS Genetics*
data availability policy, and numerical data that underlies graphs or summary statistics should be provided in spreadsheet form as supporting information., and numerical data that underlies graphs or summary statistics should be provided in spreadsheet form as supporting information., and numerical data that underlies graphs or summary statistics should be provided in spreadsheet form as supporting information., and numerical data that underlies graphs or summary statistics should be provided in spreadsheet form as supporting information.

Reviewer #1: Yes

Reviewer #2: Yes

PLOS authors have the option to publish the peer review history of their article (what does this mean?). If published, this will include your full peer review and any attached files.). If published, this will include your full peer review and any attached files.). If published, this will include your full peer review and any attached files.). If published, this will include your full peer review and any attached files.

...

Reviewer #1: No

Reviewer #2: No

**Data Deposition**

If you have submitted a Research Article or Front Matter that has associated data that are not suitable for deposition in a subject-specific public repository (such as GenBank or ArrayExpress), one way to make that data available is to deposit it in the Dryad Digital Repository. As you may recall, we ask all authors to agree to make data available; this is one way to achieve that. A full list of recommended repositories can be found on our . As you may recall, we ask all authors to agree to make data available; this is one way to achieve that. A full list of recommended repositories can be found on our . As you may recall, we ask all authors to agree to make data available; this is one way to achieve that. A full list of recommended repositories can be found on our . As you may recall, we ask all authors to agree to make data available; this is one way to achieve that. A full list of recommended repositories can be found on our website....

http://datadryad.org/submit?journalID=pgenetics&manu=PGENETICS-D-26-00149R1

Additionally, please be aware that our data availability policy requires that all numerical data underlying display items are included with the submission, and you will need to provide this before we can formally accept your manuscript, if not already present. requires that all numerical data underlying display items are included with the submission, and you will need to provide this before we can formally accept your manuscript, if not already present. requires that all numerical data underlying display items are included with the submission, and you will need to provide this before we can formally accept your manuscript, if not already present. requires that all numerical data underlying display items are included with the submission, and you will need to provide this before we can formally accept your manuscript, if not already present.

**Press Queries**

If you or your institution will be preparing press materials for this manuscript, or if you need to know your paper's publication date for media purposes, please inform the journal staff as soon as possible so that your submission can be scheduled accordingly. Your manuscript will remain under a strict press embargo until the publication date and time. This means an early version of your manuscript will not be published ahead of your final version. PLOS Genetics may also choose to issue a press release for your article. If there's anything the journal should know or you'd like more information, please get in touch via plosgenetics@plos.org....

---

## [Editor Report · Acceptance letter]

PGENETICS-D-26-00149R1

Oocyte vitrification disrupts zygotic genome activation in embryos by impairing maternal spliceosome translation and Crxos  splicing

Dear Dr Zhou,

We are pleased to inform you that your manuscript entitled "Oocyte vitrification disrupts zygotic genome activation in embryos by impairing maternal spliceosome translation and Crxos  splicing" has been formally accepted for publication in PLOS Genetics! Your manuscript is now with our production department and you will be notified of the publication date in due course.

With kind regards,

Anita Estes

PLOS Genetics

On behalf of:
